# Towards Robust Detection of Adversarial Examples

**Tianyu Pang, Chao Du, Yinpeng Dong, Jun Zhu**[*]
Dept. of Comp. Sci. & Tech., State Key Lab for Intell. Tech. & Systems
BNRist Center, THBI Lab, Tsinghua University, Beijing, China
{pty17, du-c14, dyp17}@mails.tsinghua.edu.cn, dcszj@mail.tsinghua.edu.cn

## Abstract

Although the recent progress is substantial, deep learning methods can be vulnerable to the maliciously generated adversarial examples. In this paper, we present a novel training procedure and a thresholding test strategy, towards robust detection of adversarial examples. In training, we propose to minimize the reverse cross-entropy (RCE), which encourages a deep network to learn latent representations that better distinguish adversarial examples from normal ones. In testing, we propose to use a thresholding strategy as the detector to filter out adversarial examples for reliable predictions. Our method is simple to implement using standard algorithms, with little extra training cost compared to the common cross-entropy minimization. We apply our method to defend various attacking methods on the widely used MNIST and CIFAR-10 datasets, and achieve significant improvements on robust predictions under all the threat models in the adversarial setting.

## 1 Introduction

Deep learning (DL) has obtained unprecedented progress in various tasks, including image classification, speech recognition, and natural language processing [11]. However, a high-accuracy DL model can be vulnerable in the adversarial setting [12, 33], where adversarial examples are maliciously generated to mislead the model to output wrong predictions. Several attacking methods have been developed to craft such adversarial examples [2, 4, 12, 18, 22, 29, 30]. As DL is becoming ever more prevalent, it is imperative to improve the robustness, especially in safety-critical applications.

Therefore, various defenses have been proposed attempting to correctly classify adversarial examples [14, 25, 31, 32, 33, 36]. However, most of these defenses are not effective enough, which can be successfully attacked by more powerful adversaries [2, 3]. There is also new work on verification and training provably robust networks [5, 6, 35], but these methods can only provide pointwise guarantees, and they require large extra computation cost. Overall, as adversarial examples even exist for simple classification tasks [9] and for human eyes [7], it is unlikely for such methods to solve the problem by preventing adversaries from generating adversarial examples.

Due to the difficulty, detection-based defenses have attracted a lot of attention recently as alternative solutions. Grosse et al. [13] introduce an extra class in classifiers solely for adversarial examples, and similarly Gong et al. [10] train an additional binary classifier to decide whether an instance is adversarial or not. Metzen et al. [26] detect adversarial examples via training a detection neural network, which takes input from intermediate layers of the classification network. Bhagoji et al. [1] reduce dimensionality of the input image fed to the classification network, and train a fully-connected neural network on the smaller input. Li and Li [21] build a cascade classifier where each classifier is implemented as a linear SVM acting on the PCA of inner convolutional layers of the classification network. However, these methods all require a large amount of extra computational cost, and some of them also result in loss of accuracy on normal examples. In contrast, Feinman et al. [8] propose

---

[*]Corresponding author.

a kernel density estimate method to detect the points lying far from the data manifolds in the final-layer hidden space, which does not change the structure of the classification network with little computational cost. However, Carlini and Wagner [3] show that each of these defense methods can be evaded by an adversary targeting at that specific defense, i.e., by a white-box adversary.

In this paper, we propose a defense method which consists of a novel training procedure and a thresholding test strategy. The thresholding test strategy is implemented by the kernel density (K-density) detector introduced in [8]. In training, we make contributions by presenting a novel training objective function, named as reverse cross-entropy (RCE), to substitute the common cross-entropy (CE) loss [11]. By minimizing RCE, our training procedure encourages the classifiers to return a high confidence on the true class while a uniform distribution on false classes for each data point, and further makes the classifiers map the normal examples to the neighborhood of low-dimensional manifolds in the final-layer hidden space. Compared to CE, the RCE training procedure can learn more distinguishable representations on filtering adversarial examples when using the K-density detector or other dimension-based detectors [23]. The minimization of RCE is simple to implement using stochastic gradient descent methods, with little extra training cost, as compared to CE. Therefore, it can be easily applied to any deep networks and is as scalable as the CE training procedure.

We apply our method to defend various attacking methods on the widely used MNIST [20] and CIFAR-10 [17] datasets. We test the performance of our method under different threat models, i.e., *oblivious adversaries*, *white-box adversaries* and *black-box adversaries*. We choose the K-density estimate method as our strong baseline, which has shown its superiority and versatility compared to other detection-based defenses [3]. The results demonstrate that compared to the baseline, the proposed method improves the robustness against adversarial attacks under all the threat models, while maintaining state-of-the-art accuracy on normal examples. Specifically, we demonstrate that the white-box adversaries have to craft adversarial examples with macroscopic noises to successfully evade our defense, which means human observers can easily filter out the crafted adversarial examples.

## 2 Preliminaries

This section provides the notations and introduces the threat models and attacking methods.

### 2.1 Notations

A deep neural network (DNN) classifier can be generally expressed as a mapping function $F(X, \theta) : \mathbb{R}^d \to \mathbb{R}^L$, where $X \in \mathbb{R}^d$ is the input variable, $\theta$ denotes all the parameters and $L$ is the number of classes (hereafter we will omit $\theta$ without ambiguity). Here, we focus on the DNNs with softmax output layers. For notation clarity, we define the softmax function $\mathbb{S}(z) : \mathbb{R}^L \to \mathbb{R}^L$ as $\mathbb{S}(z)_i = \exp(z_i) / \sum_{i=1}^{L} \exp(z_i), i \in [L]$, where $[L] := \{1, \cdots, L\}$. Let $Z$ be the output vector of the penultimate layer, i.e., the final hidden layer. This defines a mapping function: $X \mapsto Z$ to extract data representations. Then, the classifier can be expressed as $F(X) = \mathbb{S}(W_s Z + b_s)$, where $W_s$ and $b_s$ are the weight matrix and bias vector of the softmax layer respectively. We denote the pre-softmax output $W_s Z + b_s$ as $Z_{pre}$, termed logits. Given an input $x$ (i.e., an instance of $X$), the predicted label for $x$ is denoted as $\hat{y} = \arg\max_{i \in [L]} F(x)_i$. The probability value $F(x)_{\hat{y}}$ is often used as the confidence score on this prediction [11]. One common training objective is to minimize the cross-entropy (CE) loss, which is defined as:

$$\mathcal{L}_{CE}(x, y) = -1_y^\top \log F(x) = -\log F(x)_y,$$

for a single input-label pair $(x, y)$. Here, $1_y$ is the one-hot encoding of $y$ and the logarithm of a vector is defined as taking logarithm of each element. The CE training procedure intends to minimize the average CE loss (under proper regularization) on training data to obtain the optimal parameters.

### 2.2 Threat models

In the adversarial setting, an elaborate taxonomy of threat models is introduced in [3]:
- **Oblivious adversaries** are not aware of the existence of the detector $D$ and generate adversarial examples based on the unsecured classification model $F$.
- **White-box adversaries** know the scheme and parameters of $D$, and can design special methods to attack both the model $F$ and the detector $D$ simultaneously.
- **Black-box adversaries** know the existence of the detector $D$ with its scheme, but have no access to the parameters of the detector $D$ or the model $F$.

## 2.3 Attacking methods

Although DNNs have obtained substantial progress, adversarial examples can be easily identified to fool the network, even when its accuracy is high [28]. Several attacking methods on generating adversarial examples have been introduced in recent years. Most of them can craft adversarial examples that are visually indistinguishable from the corresponding normal ones, and yet are misclassified by the target model $F$. Here we introduce some well-known and commonly used attacking methods.

**Fast Gradient Sign Method (FGSM):** Goodfellow et al. [12] introduce an one-step attacking method, which crafts an adversarial example $x^*$ as $x^* = x + \epsilon \cdot \text{sign}(\nabla_x \mathcal{L}(x, y))$, with the perturbation $\epsilon$ and the training loss $\mathcal{L}(x, y)$.

**Basic Iterative Method (BIM):** Kurakin et al. [18] propose an iterative version of FGSM, with the formula as $x_i^* = \text{clip}_{x,\epsilon}(x_{i-1}^* + \frac{\epsilon}{r} \cdot \text{sign}(\nabla_{x_{i-1}^*} \mathcal{L}(x_{i-1}^*, y)))$, where $x_0^* = x$, $r$ is the number of iteration steps and $\text{clip}_{x,\epsilon}(\cdot)$ is a clipping function to keep $x_i^*$ in its domain.

**Iterative Least-likely Class Method (ILCM):** Kurakin et al. [18] also propose a targeted version of BIM as $x_i^* = \text{clip}_{x,\epsilon}(x_{i-1}^* - \frac{\epsilon}{r} \cdot \text{sign}(\nabla_{x_{i-1}^*} \mathcal{L}(x_{i-1}^*, y_{ll})))$, where $x_0^* = x$ and $y_{ll} = \arg\min_i F(x)_i$. ILCM can avoid label leaking [19], since it does not exploit information of the true label $y$.

**Jacobian-based Saliency Map Attack (JSMA):** Papernot et al. [30] propose another iterative method for targeted attack, which perturbs one feature $x_i$ by a constant offset $\epsilon$ in each iteration step that maximizes the saliency map

$$S(x, t)[i] = \begin{cases} 0, \text{if } \frac{\partial F(x)_y}{\partial x_i} < 0 \text{ or } \sum_{j \neq y} \frac{\partial F(x)_j}{\partial x_i} > 0, \\ (\frac{\partial F(x)_y}{\partial x_i}) \left| \sum_{j \neq y} \frac{\partial F(x)_j}{\partial x_i} \right|, \text{otherwise.} \end{cases}$$

Compared to other methods, JSMA perturbs fewer pixels.

**Carlini & Wagner (C&W):** Carlini and Wagner [2] introduce an optimization-based method, which is one of the most powerful attacks. They define $x^* = \frac{1}{2}(\tanh(\omega) + 1)$ in terms of an auxiliary variable $\omega$, and solve the problem $\min_\omega \|\frac{1}{2}(\tanh(\omega) + 1) - x\|_2^2 + c \cdot f(\frac{1}{2}(\tanh(\omega) + 1))$, where $c$ is a constant that need to be chosen by modified binary search. $f(\cdot)$ is an objective function as $f(x) = \max(\max\{Z_{pre}(x)_i : i \neq y\} - Z_{pre}(x)_i, -\kappa)$, where $\kappa$ controls the confidence.

# 3 Methodology

In this section, we present a new method to improve the robustness of classifiers in the adversarial setting. We first construct a new metric and analyze its properties, which guides us to the new method.

## 3.1 Non-maximal entropy

Due to the difficulty of correctly classifying adversarial examples [2, 3] and the generality of their existence [7, 9], we design a method to detect them instead, which could help in real world applications. For example, in semi-autonomous systems, the detection of adversarial examples would allow disabling autonomous operation and requesting human intervention [26].

A detection method relies on some metrics to decide whether an input $x$ is adversarial or not for a given classifier $F(X)$. A potential candidate is the confidence $F(x)_{\hat{y}}$ on the predicted label $\hat{y}$, which inherently conveys the degree of certainty on a prediction and is widely used [11]. However, previous work shows that the confidence score is unreliable in the adversarial setting [12, 28]. Therefore, we construct another metric which is more pertinent and helpful to our goal. Namely, we define the metric of *non-ME*—the entropy of normalized non-maximal elements in $F(x)$, as:

$$\text{non-ME}(x) = -\sum_{i \neq \hat{y}} \hat{F}(x)_i \log(\hat{F}(x)_i), \tag{1}$$

where $\hat{F}(x)_i = F(x)_i / \sum_{j \neq \hat{y}} F(x)_j$ are the normalized non-maximal elements in $F(x)$. Hereafter we will consider the final hidden vector $z$ of $F$ given $x$, and use the notation $F(z)$ with the same meaning as $F(x)$ without ambiguity. To intuitively illustrate the ideas, Fig. 1a presents an example of classifier $F$ in the hidden space, where $z \in \mathbb{R}^2$ and $L = 3$. Let $Z_{pre,i}, i \in [L]$ be the $i$-th element of the logits $Z_{pre}$. Then the decision boundary between each pair of classes $i$ and $j$ is the hyperplane $db_{ij} = \{z : Z_{pre,i} = Z_{pre,j}\}$, and let $DB_{ij} = \{Z_{pre,i} = Z_{pre,j} + C, C \in \mathbb{R}\}$ be the set of all

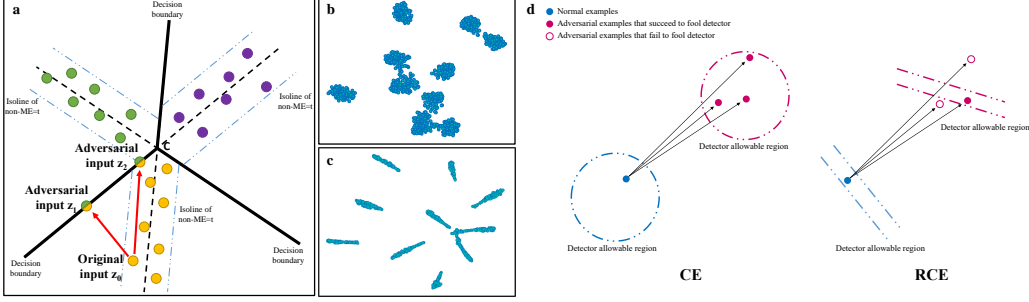

Figure 1: **a**, The three black solid lines are the decision boundary of the classifier, and each black line (both solid and dashed parts) is the decision boundary between two classes. The blue dot-dashed lines are the isolines of non-ME $= t$. **b**, t-SNE visualization of the final hidden vectors on CIFAR-10. The model is Resnet-32. The training procedure is CE. **c**, The training procedure is RCE. **d**, Practical attacks on the trained networks. Blue regions are of the original classes for normal examples, and red regions are of the target classes for adversarial ones.

parallel hyperplanes w.r.t. $db_{ij}$. In Fig. 1a, each $db_{ij}$ corresponds to one of the three black lines. We denote the half space $Z_{pre,i} \geq Z_{pre,j}$ as $db_{ij}^+$. Then, we can formally represent the decision region of class $\hat{y}$ as $dd_{\hat{y}} = \bigcap_{i \neq \hat{y}} db_{\hat{y}i}^+$ and the corresponding decision boundary of this region as $\overline{dd_{\hat{y}}}$. Note that the output $F(z)$ has $L-1$ equal non-maximal elements for any points on the low-dimensional manifold $S_{\hat{y}} = (\bigcap_{i,j \neq \hat{y}} db_{ij}) \bigcap dd_{\hat{y}}$. With the above notations, we have Lemma 1 as below:

**Lemma 1.** *(Proof in Appendix A) In the decision region $dd_{\hat{y}}$ of class $\hat{y}$, $\forall i, j \neq \hat{y}, \widetilde{db_{ij}} \in DB_{ij}$, the value of* non-ME *for any point on the low-dimensional manifold $\bigcap_{i,j \neq \hat{y}} \widetilde{db_{ij}}$ is constant. In particular,* non-ME *obtains its global maximal value $\log(L-1)$ on and only on $S_{\hat{y}}$.*

Lemma 1 tells us that in the decision region of class $\hat{y}$, if one moves a normal input along the low-dimensional manifold $\bigcap_{i,j \neq \hat{y}} \widetilde{db_{ij}}$, then its value of non-ME will not change, and vice verse.

**Theorem 1.** *(Proof in Appendix A) In the decision region $dd_{\hat{y}}$ of class $\hat{y}$, $\forall i, j \neq \hat{y}, z_0 \in dd_{\hat{y}}$, there exists a unique $\widetilde{db_{ij}^0} \in DB_{ij}$, such that $z_0 \in Q_0$, where $Q_0 = \bigcap_{i,j \neq \hat{y}} \widetilde{db_{ij}^0}$. Let $Q_0^{\hat{y}} = Q_0 \bigcap \overline{dd_{\hat{y}}}$, then the solution set of the problem*

$$\arg \min_{z_0} (\max_{z \in Q_0^{\hat{y}}} F(z)_{\hat{y}})$$

*is $S_{\hat{y}}$. Furthermore, $\forall z_0 \in S_{\hat{y}}$ there is $Q_0 = S_{\hat{y}}$, and $\forall z \in S_{\hat{y}} \bigcap \overline{dd_{\hat{y}}}$, $F(z)_{\hat{y}} = \frac{1}{L}$.*

Let $z_0$ be the representation of a normal example with the predicted class $\hat{y}$. When crafting adversarial examples based on $z_0$, adversaries need to perturb $z_0$ across the decision boundary $\overline{dd_{\hat{y}}}$. Theorem 1 says that there exists a unique low-dimensional manifold $Q_0$ that $z_0$ lies on in the decision region of class $\hat{y}$. If we can somehow restrict adversaries changing the values of non-ME when they perturb $z_0$, then by Lemma 1, the adversaries can only perturb $z_0$ along the manifold $Q_0$. In this case, the nearest adversarial counterpart $z^*$ for $z_0$ must be in the set $Q_0^{\hat{y}}$ [27]. Then the value of $\max_{z \in Q_0^{\hat{y}}} F(z)_{\hat{y}}$ is an upper bound of the prediction confidence $F(z^*)_{\hat{y}}$. This bound is a function of $z_0$. Theorem 1 further tells us that if $z_0 \in S_{\hat{y}}$, the corresponding value of the upper bound will obtain its minimum $\frac{1}{L}$, which leads to $F(z^*)_{\hat{y}} = \frac{1}{L}$. This makes $z^*$ be easily distinguished since its low confidence score.

In practice, the restriction can be implemented by a detector with the metric of non-ME. In the case of Fig. 1a, any point that locates on the set $S_{\hat{y}}$ (black dashed lines) has the highest value of non-ME $= \log 2$. Assuming that the learned representation transformation: $X \mapsto Z$ can map all the normal examples to the neighborhood of $S_{\hat{y}}$, where the neighborhood boundary consists of the isolines of non-ME $= t$ (blue dot-dashed lines). This means that all the normal examples have values of non-ME $> t$. When there is no detector, the nearest adversarial example based on $z_0$ is $z_1$, which locates on the nearest decision boundary w.r.t. $z_0$. In contrast, when non-ME is used as the detection metric, $z_1$ will be easily filtered out by the detector because non-ME$(z_1) < t$, and the nearest adversarial example becomes $z_2$ in this case, which locates on the junction manifold of the neighborhood boundary and the decision boundary. It is easy to generally conclude that $\|z_0 - z_2\| > \|z_0 - z_1\|$, almost everywhere. This means that due to the existence of the detector, adversaries have to impose larger minimal perturbations to successfully generate adversarial examples

that can fool the detector. Furthermore, according to Theorem 1, the confidence at $z_2$ is also lower than it at $z_1$, which makes $z_2$ still be most likely be distinguished since its low confidence score.

## 3.2 The reverse cross-entropy training procedure

Based on the above analysis, we now design a new training objective to improve the robustness of DNN classifiers. The key is to enforce a DNN classifier to map all the normal instances to the neighborhood of the low-dimensional manifolds $S_{\hat{y}}$ in the final-layer hidden space. According to Lemma 1, this can be achieved by making the non-maximal elements of $F(x)$ be as equal as possible, thus having a high non-ME value for every normal input. Specifically, for a training data $(x, y)$, we let $R_y$ denote its reverse label vector whose $y$-th element is zero and other elements equal to $\frac{1}{L-1}$. One obvious way to encourage uniformity among the non-maximal elements of $F(x)$ is to apply the model regularization method termed label smoothing [34], which can be done by introducing a cross-entropy term between $R_y$ and $F(x)$ in the CE objective:

$$\mathcal{L}^{\lambda}_{CE}(x, y) = \mathcal{L}_{CE}(x, y) - \lambda \cdot R_y^{\top} \log F(x), \tag{2}$$

where $\lambda$ is a trade-off parameter. However, it is easy to show that minimizing $\mathcal{L}^{\lambda}_{CE}$ equals to minimizing the cross-entropy between $F(x)$ and the $L$-dimensional vector $P^{\lambda}$:

$$P_i^{\lambda} = \begin{cases} \frac{1}{\lambda+1}, & i = y, \\ \frac{\lambda}{(L-1)(\lambda+1)}, & i \neq y. \end{cases} \tag{3}$$

Note that $1_y = P^0$ and $R_y = P^{\infty}$. When $\lambda > 0$, let $\theta^*_{\lambda} = \arg\min_{\theta} \mathcal{L}^{\lambda}_{CE}$, then the prediction $F(x, \theta^*_{\lambda})$ will tend to equal to $P^{\lambda}$, rather than the ground-truth $1_y$. This makes the output predictions be biased. In order to have unbiased predictions that make the output vector $F(x)$ tend to $1_y$, and simultaneously encourage uniformity among probabilities on untrue classes, we define another objective function based on what we call *reverse cross-entropy (RCE)* as

$$\mathcal{L}^R_{CE}(x, y) = -R_y^{\top} \log F(x). \tag{4}$$

Minimizing RCE is equivalent to minimizing $\mathcal{L}^{\infty}_{CE}$. Note that by directly minimizing $\mathcal{L}^R_{CE}$, i.e., $\theta^*_R = \arg\min_{\theta} \mathcal{L}^R_{CE}$, one will get a reverse classifier $F(X, \theta^*_R)$, which means that given an input $x$, the reverse classifier $F(X, \theta^*_R)$ will not only tend to assign the lowest probability to the true class but also tend to output a uniform distribution on other classes. This simple insight leads to our entire RCE training procedure which consists of two parts, as outlined below:

**Reverse training:** Given the training set $\mathcal{D} := \{(x^i, y^i)\}_{i \in [N]}$, training the DNN $F(X, \theta)$ to be a reverse classifier by minimizing the average RCE loss: $\theta^*_R = \arg\min_{\theta} \frac{1}{N} \sum_{i=1}^N \mathcal{L}^R_{CE}(x^i, y^i)$.

**Reverse logits:** Negating the final logits fed to the softmax layer as $F_R(X, \theta^*_R) = \mathbb{S}(-Z_{pre}(X, \theta^*_R))$.

Then we will obtain the network $F_R(X, \theta^*_R)$ that returns ordinary predictions on classes, and $F_R(X, \theta^*_R)$ is referred as the network trained via *the RCE training procedure*.

**Theorem 2.** *(Proof in Appendix A) Let $(x, y)$ be a given training data. Under the $L_{\infty}$-norm, if there is a training error $\alpha \ll \frac{1}{L}$ that $\|\mathbb{S}(Z_{pre}(x, \theta^*_R)) - R_y\|_{\infty} \leq \alpha$, then we have bounds*

$$\|\mathbb{S}(-Z_{pre}(x, \theta^*_R)) - 1_y\|_{\infty} \leq \alpha(L-1)^2,$$

*and $\forall j, k \neq y$,*

$$|\mathbb{S}(-Z_{pre}(x, \theta^*_R))_j - \mathbb{S}(-Z_{pre}(x, \theta^*_R))_k| \leq 2\alpha^2(L-1)^2.$$

Theorem 2 demonstrates two important properties of the RCE training procedure. First, it is consistent and unbiased that when the training error $\alpha \to 0$, the output $F_R(x, \theta^*_R)$ converges to the one-hot label vector $1_y$. Second, the upper bounds of the difference between any two non-maximal elements in outputs decrease as $\mathcal{O}(\alpha^2)$ w.r.t. $\alpha$ for RCE, much faster than the $\mathcal{O}(\alpha)$ for CE and label smoothing. These two properties make the RCE training procedure meet our requirements as described above.

## 3.3 The thresholding test strategy

Given a trained classifier $F(X)$, we implement a thresholding test strategy by a detector for robust prediction. After presetting a metric, the detector classifies the input as normal and decides to return

the predicted label if the value of metric is larger than a threshold $T$, or classifies the one as adversarial and returns NOT SURE otherwise. In our method, we adopt the kernel density (K-density) metric introduced in [8], because applying the K-density metric with CE training has already shown better robustness and versatility than other defenses [3]. K-density can be regarded as some combination of the confidence and non-ME metrics, since it can simultaneously convey the information about them.

**Kernel density:** The K-density is calculated in the final-layer hidden space. Given the predicted label $\hat{y}$, K-density is defined as $KD(x) = \frac{1}{|X_{\hat{y}}|} \sum_{x_i \in X_{\hat{y}}} k(z_i, z)$, where $X_{\hat{y}}$ represents the set of training points with label $\hat{y}$, $z_i$ and $z$ are the corresponding final-layer hidden vectors, $k(z_i, z) = \exp(-\|z_i - z\|^2/\sigma^2)$ is the Gaussian kernel with the bandwidth $\sigma$ treated as a hyperparameter.

Carlini and Wagner [3] show that previous methods on detecting adversarial examples can be evaded by white-box adversaries. However, our method (RCE training + K-density detector) can defend the white-box attacks effectively. This is because the RCE training procedure conceals normal examples on low-dimensional manifolds in the final-layer hidden space, as shown in Fig. 1b and Fig. 1c. Then the detector allowable regions can also be set low-dimensional as long as the regions contain all normal examples. Therefore the white-box adversaries who intend to fool our detector have to generate adversarial examples with preciser calculations and larger noises. This is intuitively illustrated in Fig. 1d, where the adversarial examples crafted on the networks trained by CE are easier to locate in the detector allowable regions than those crafted on the networks trained by RCE. This illustration is experimentally verified in Section 4.4.

## 4 Experiments

We now present the experimental results to demonstrate the effectiveness of our method on improving the robustness of DNN classifiers in the adversarial setting.

### 4.1 Setup

We use the two widely studied datasets—MNIST [20] and CIFAR-10 [17]. MNIST is a collection of handwritten digits with a training set of 60,000 images and a test set of 10,000 images. CIFAR-10 consists of 60,000 color images in 10 classes with 6,000 images per class. There are 50,000 training images and 10,000 test images. The pixel values of images in both datasets are scaled to be in the interval $[-0.5, 0.5]$. The normal examples in our experiments refer to all the ones in the training and test sets. In the adversarial setting, the *strong baseline* we use is the K-density estimate method (CE training + K-density detector) [8], which has shown its superiority and versatility compared to other detection-based defense methods [1, 10, 13, 21, 26] in [3].

### 4.2 Classification on normal examples

We first evaluate in the normal setting, where we implement Resnet-32 and Resnet-56 [15] on both datasets. For each network, we use both the CE and RCE as the training objectives, trained by the same settings as He et al. [16]. The number of training steps for both objectives is set to be 20,000 on MNIST and 90,000 on CIFAR-10. Hereafter for notation simplicity, we will indicate the

Table 1: Classification error rates (%) on test sets.

| Method | MNIST | CIFAR-10 |
|---|---|---|
| Resnet-32 (CE) | 0.38 | 7.13 |
| Resnet-32 (RCE) | **0.29** | **7.02** |
| Resnet-56 (CE) | 0.36 | **6.49** |
| Resnet-56 (RCE) | **0.32** | 6.60 |

training procedure used after the model name of a trained network, e.g., Resnet-32 (CE). Similarly, we indicate the training procedure and omit the name of the target network after an attacking method, e.g., FGSM (CE). Table 1 shows the test error rates, where the thresholding test strategy is disabled and all the points receive their predicted labels. We can see that the performance of the networks trained by RCE is as good as and sometimes even better than those trained by the traditional CE procedure. Note that we apply the same training hyperparameters (e.g., learning rates and decay factors) for both the CE and RCE procedures, which suggests that RCE is easy to optimize and does not require much extra effort on tuning hyperparameters.

To verify that the RCE procedure tends to map all the normal inputs to the neighborhood of $S_{\hat{y}}$ in the hidden space, we apply the t-SNE technique [24] to visualize the distribution of the final hidden vector $z$ on the test set. Fig. 1b and Fig. 1c give the 2-D visualization results on 1,000 test examples of CIFAR-10. We can see that the networks trained by RCE can successfully map the test examples to the neighborhood of low-dimensional manifolds in the final-layer hidden space.

Table 2: AUC-scores $(10^{-2})$ of adversarial examples. The model of target networks is Resnet-32. Values are calculated on the examples which are correctly classified as normal examples and then misclassified as adversarial counterparts. Bandwidths used when calculating K-density are $\sigma^2_{CE} = 1/0.26$ and $\sigma^2_{RCE} = 0.1/0.26$. *Here (-) indicates the strong baseline, and (*) indicates our defense method.*

| Attack | Obj. | MNIST | | | CIFAR-10 | | |
|--------|------|-------|--------|-----------|----------|--------|-----------|
| | | Confidence | non-ME | K-density | Confidence | non-ME | K-density |
| FGSM | CE | 79.7 | 66.8 | 98.8 (-) | 71.5 | 66.9 | **99.7** (-) |
| | RCE | 98.8 | 98.6 | **99.4** (*) | 92.6 | 91.4 | 98.0 (*) |
| BIM | CE | 88.9 | 70.5 | 90.0 (-) | 0.0 | 64.6 | **100.0** (-) |
| | RCE | 91.7 | 90.6 | **91.8** (*) | 0.7 | 70.2 | **100.0** (*) |
| ILCM | CE | 98.4 | 50.4 | 96.2 (-) | 16.4 | 37.1 | 84.2 (-) |
| | RCE | 100.0 | 97.0 | **98.6** (*) | 64.1 | 77.8 | **93.9** (*) |
| JSMA | CE | 98.6 | 60.1 | 97.7 (-) | 99.2 | 27.3 | 85.8 (-) |
| | RCE | 100.0 | 99.4 | **99.0** (*) | 99.5 | 91.9 | **95.4** (*) |
| C&W | CE | 98.6 | 64.1 | 99.4 (-) | 99.5 | 50.2 | 95.3 (-) |
| | RCE | 100.0 | 99.5 | **99.8** (*) | 99.6 | 94.7 | **98.2** (*) |
| C&W-hc | CE | 0.0 | 40.0 | 91.1 (-) | 0.0 | 28.8 | 75.4 (-) |
| | RCE | 0.1 | 93.4 | **99.6** (*) | 0.2 | 53.6 | **91.8** (*) |

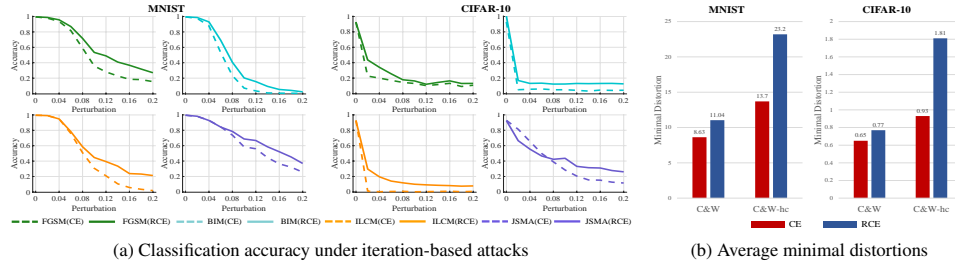

(a) Classification accuracy under iteration-based attacks      (b) Average minimal distortions

Figure 2: Robustness with the thresholding test strategy disabled. The model of target networks is Resnet-32.

## 4.3 Performance under the oblivious attack

We test the performance of the trained Resnet-32 networks on MNIST and CIFAR-10 under the oblivious attack, where we investigate the attacking methods as in Sec. 2.3. We first disable the thresholding test strategy and make classifiers return all predictions to study the networks ability of correctly classifying adversarial examples. We use the iteration-based attacking methods: FGSM, BIM, ILCM and JSMA, and calculate the classification accuracy of networks on crafted adversarial examples w.r.t. the perturbation $\epsilon$. Fig. 2a shows the results. We can see that Resnet-32 (RCE) has higher accuracy scores than Resnet-32 (CE) under all the four attacks on both datasets.

As for optimization-based methods like the C&W attack and its variants, we apply the same way as in [3] to report robustness. Specifically, we do a binary search for the parameter $c$, in order to find the minimal distortions that can successfully attack the classifier. The distortion is defined in [33] as $\|x - x^*\|_2 / \sqrt{d}$, where $x^*$ is the generated adversarial example and each pixel feature is rescaled to be in the interval $[0, 255]$. We set the step size in the C&W attacks at 0.01, and set binary search rounds of $c$ to be 9 with the maximal iteration steps at 10,000 in each round. Moreover, to make our investigation more convincing, we introduce the high-confidence version of the C&W attack (abbr. to C&W-hc) that sets the parameter $\kappa$ in the C&W attack to be 10 in our experiments. The C&W-hc attack can generate adversarial examples with the confidence higher than 0.99, and previous work has shown that the adversarial examples crafted by C&W-hc are stronger and more difficult to defend than those crafted by C&W [2, 3]. The results are shown in Fig. 2b. We can see that the C&W and C&W-hc attacks need much larger minimal distortions to successfully attack the networks trained by RCE than those trained by CE. Similar phenomenon is also observed under the white-box attack.

We further activate the thresholding test strategy with the K-density metric, and also test the performance of confidence or non-ME being the metric for a more complete analysis. We construct simple binary classifiers to decide whether an example is adversarial or not by thresholding with the metrics, and then calculate the AUC-scores of ROC curves on these binary classifiers. Table 2 shows the AUC-scores calculated under different combinations of training procedures and thresholding metrics on both datasets. From Table 2, we can see that our method (RCE training + K-density detector) performs the best in almost all cases, and non-ME itself is also a pretty reliable metric, although not as good as K-density. The classifiers trained by RCE also return more reliable confidence scores, which verifies the conclusion in Theorem 1. Furthermore, we also show that our method can better distinguish between noisy examples and adversarial examples, as demonstrated in Appendix B.3.

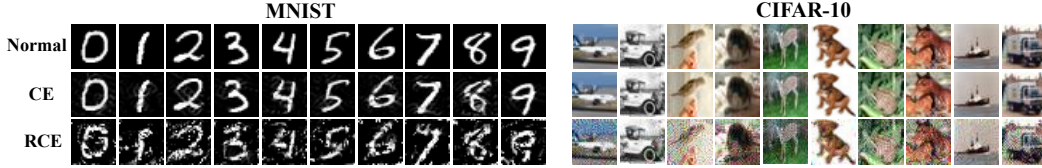

Figure 3: The normal test images are termed as *Normal*, and adversarial examples generated on Resnet-32 (CE) and Resnet-32 (RCE) are separately termed as *CE / RCE*. Adversarial examples are generated by C&W-wb with minimal distortions.

| Obj. | MNIST | | CIFAR-10 | |
|---|---|---|---|---|
| | Ratio | Distortion | Ratio | Distortion |
| CE | 0.01 | 17.12 | 0.00 | 1.26 |
| RCE | **0.77** | **31.59** | **0.12** | **3.89** |

Table 3: The ratios of $f_2(x^*) > 0$ and minimal distortions of the adversarial examples crafted by C&W-wb. Model is Resnet-32.

| | Res.-32 (CE) | Res.-32 (RCE) |
|---|---|---|
| Res.-56 (CE) | 75.0 | 90.8 |
| Res.-56 (RCE) | 89.1 | 84.9 |

Table 4: AUC-scores $(10^{-2})$ on CIFAR-10. Resnet-32 is the substitute model and Resnet-56 is the target model.

## 4.4 Performance under the white-box attack

We test our method under the white-box attack, which is the most difficult threat model and no effective defense exits yet. We apply the white-box version of the C&W attack (abbr. to C&W-wb) introduced in [3], which is constructed specially to fool the K-density detectors. C&W-wb is also a white-box attack for our method, since it does not exploit information of the training objective. C&W-wb introduces a new loss term $f_2(x^*) = \max(-\log(KD(x^*)) - \eta, 0)$ that penalizes the adversarial example $x^*$ being detected by the K-density detectors, where $\eta$ is set to be the median of $-\log(KD(\cdot))$ on the training set. Table 3 shows the average minimal distortions and the ratios of $f_2(x^*) > 0$ on the adversarial examples crafted by C&W-wb, where a higher ratio indicates that the detector is more robust and harder to be fooled. We find that nearly all the adversarial examples generated on Resnet-32 (CE) have $f_2(x^*) \leq 0$, which means that the values of K-density on them are greater than half of the values on the training data. This result is consistent with previous work [3].

However, note that applying C&W-wb on our method has a much higher ratio and results in a much larger minimal distortion. Fig. 3 shows some adversarial examples crafted by C&W-wb with the corresponding normal ones. We find that the adversarial examples crafted on Resnet-32 (CE) are indistinguishable from the normal ones by human eyes. In contrast, those crafted on Resnet-32 (RCE) have macroscopic noises, which are not strictly adversarial examples since they are visually distinguishable from normal ones. The inefficiency of the most aggressive attack C&W-wb under our defense verifies our illustration in Fig. 1d. More details on the limitation of C&W-wb are in Appendix B.4. We have also designed white-box attacks that exploit the training loss information of RCE, and we get inefficient attacks compared to C&W-wb. This is because given an input, there is no explicit relationship between its RCE value and K-density score. Thus it is more efficient to directly attack the K-density detectors as C&W-wb does.

## 4.5 Performance under the black-box attack

For complete analysis, we investigate the robustness under the black-box attack. The success of the black-box attack is based on the transferability of adversarial examples among different models [12]. We set the trained Resnet-56 networks as the target models. Adversaries intend to attack them but do not have access to their parameters. Thus we set the trained Resnet-32 networks to be the substitute models that adversaries actually attack on and then feed the crafted adversarial examples into the target models. Since adversaries know the existence of the K-density detectors, we apply the C&W-wb attack. We find that the adversarial examples crafted by the C&W-wb attack have poor transferability, where less than 50% of them can make the target model misclassify on MNIST and less than 15% on CIFAR-10. Table 4 shows the AUC-scores in four different cases of the black-box attack on CIFAR-10, and the AUC-scores in the same cases on MNIST are all higher than 95%. Note that in our experiments the target models and the substitute models have very similar structures, and the C&W-wb attack becomes ineffective even under this quite 'white' black-box attack.

## 5 Conclusions

We present a novel method to improve the robustness of deep learning models by reliably detecting and filtering out adversarial examples, which can be implemented using standard algorithms with little extra training cost. Our method performs well on both the MNIST and CIFAR-10 datasets under all threat models and various attacking methods, while maintaining accuracy on normal examples.

## Acknowledgements

This work was supported by the National Key Research and Development Program of China (No. 2017YFA0700904), NSFC Projects (Nos. 61620106010, 61621136008, 61332007), Beijing NSF Project (No. L172037), Tiangong Institute for Intelligent Computing, NVIDIA NVAIL Program, and the projects from Siemens and Intel.

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
