[Supplementary Material]

# Appendix

## A  Proof

**Lemma 1.** *In the decision region $dd_{\hat{y}}$ of class $\hat{y}$, $\forall i, j \neq \hat{y}, \widetilde{db_{ij}} \in DB_{ij}$, the value of* non-ME *for any point on the low-dimensional manifold $\bigcap_{i,j \neq \hat{y}} \widetilde{db_{ij}}$ is constant. In particular,* non-ME *obtains its global maximal value $\log(L-1)$ on and only on $S_{\hat{y}}$.*

*Proof.* $\forall i, j \neq \hat{y}$, we take a hyperplane $\widetilde{db_{ij}} \in DB_{ij}$. Then according to the definition of the set $DB_{ij}$, it is easily shown that $\forall z \in \widetilde{db_{ij}}$, $Z_{pre,i} - Z_{pre,j} = constant$, and we denote this corresponding constant as $C_{ij}$. Thus given any $k \neq \hat{y}$, we derive that $\forall z \in \bigcap_{i,j \neq \hat{y}} \widetilde{db_{ij}}$

$$
\begin{aligned}
\hat{F}(z)_k &= \frac{F(z)_k}{\sum_{j \neq \hat{y}} F(z)_j} \\
&= \frac{\exp(Z_{pre,k})}{\sum_{j \neq \hat{y}} \exp(Z_{pre,j})} \\
&= \frac{1}{\sum_{j \neq \hat{y}} \exp(Z_{pre,j} - Z_{pre,k})} \\
&= \frac{1}{\sum_{j \neq \hat{y}} \exp(C_{jk})} \\
&= constant,
\end{aligned}
$$

and according to the defination of the non-ME value $\text{non-ME}(z) = -\sum_{i \neq \hat{y}} \hat{F}(z)_i \log(\hat{F}(z)_i)$, we can conclude that $\text{non-ME}(z) = constant, \forall z \in \bigcap_{i,j \neq \hat{y}} \widetilde{db_{ij}}$.

In particular, according to the property of entropy in information theory, we know that $\text{non-ME} \leq \log(L-1)$, and non-ME achieve its maximal value if and only if $\forall k \neq \hat{y}, \hat{F}_k = \frac{1}{L-1}$. In this case, there is $\forall i, j \neq \hat{y}, Z_{pre,i} = Z_{pre,j}$, which is easy to show that the conditions hold on $S_{\hat{y}}$. Conversely, $\forall z \notin S_{\hat{y}}$, there must $\exists i, j \neq \hat{y}$, such that $Z_{pre,i} \neq Z_{pre,j}$ which leads to $\hat{F}_i \neq \hat{F}_j$. This violates the condition of non-ME achieving its maximal value. Thus non-ME obtains its global maximal value $\log(L-1)$ on and only on $S_{\hat{y}}$.

$\square$

**Theorem 1.** *In the decision region $dd_{\hat{y}}$ of class $\hat{y}$, $\forall i, j \neq \hat{y}, z_0 \in dd_{\hat{y}}$, there exists a unique $\widetilde{db_{ij}^0} \in DB_{ij}$, such that $z_0 \in Q_0$, where $Q_0 = \bigcap_{i,j \neq \hat{y}} \widetilde{db_{ij}^0}$. Let $Q_0^{\hat{y}} = Q_0 \bigcap \overline{dd_{\hat{y}}}$, then the solution set of the problem*

$$\arg\min_{z_0} (\max_{z^* \in Q_0^{\hat{y}}} F(z^*)_{\hat{y}})$$

*is $S_{\hat{y}}$. Furthermore, $\forall z_0 \in S_{\hat{y}}$ there is $Q_0 = S_{\hat{y}}$, and $\forall z^* \in S_{\hat{y}} \bigcap \overline{dd_{\hat{y}}}$, $F(z^*)_{\hat{y}} = \frac{1}{L}$.*

*Proof.* It is easy to show that given a point and a normal vector, one can uniquely determine a hyperplane. Thus $\forall i, j \neq \hat{y}, z_0 \in dd_{\hat{y}}$, there exists unique $\widetilde{db_{ij}^0} \in DB_{ij}$, such that $z_0 \in \bigcap_{i,j \neq \hat{y}} \widetilde{db_{ij}^0} = Q_0$.

According to the proof of Lemma 1, we have $\forall i, j \neq \hat{y}, z^* \in Q_0^{\hat{y}}$, there is $Z_{pre,i} - Z_{pre,j} = C_{ij}$, and $\exists k \neq \hat{y}, \text{s. t. } Z_{pre,\hat{y}} = Z_{pre,k}$. Thus we can derive

$$
\begin{aligned}
F(z^*)_{\hat{y}} &= \frac{\exp(Z_{pre,\hat{y}})}{\sum_i \exp(Z_{pre,i})} \\
&= \frac{1}{1 + \sum_{i \neq \hat{y}} \exp(Z_{pre,i} - Z_{pre,\hat{y}})} \\
&= \frac{1}{1 + \exp(Z_{pre,k} - Z_{pre,\hat{y}})(1 + \sum_{i \neq \hat{y},k} \exp(Z_{pre,i} - Z_{pre,k}))} \\
&= \frac{1}{2 + \sum_{i \neq \hat{y},k} \exp(C_{ik})}.
\end{aligned}
$$

Let $M = \{i : C_{ij} \geq 0, \forall j \neq \hat{y}\}$, there must be $k \in M$ so $M$ is not empty, and we have

$$
\begin{aligned}
\max_{z^* \in Q_0^{\hat{y}}} F(z^*)_{\hat{y}} &= \max_{l \in M} \frac{1}{2 + \sum_{i \neq \hat{y},l} \exp(C_{il})} \\
&= \frac{1}{2 + \min_{l \in M} \sum_{i \neq \hat{y},l} \exp(C_{il})} \\
&= \frac{1}{2 + \sum_{i \neq \hat{y},\widetilde{k}} \exp(C_{i\widetilde{k}})},
\end{aligned}
$$

where $\widetilde{k}$ is any element in $M$. This equation holds since $\forall k_1, k_2 \in M$, there is $C_{k_1 k_2} \geq 0, C_{k_2 k_1} \geq 0$ and $C_{k_1 k_2} = -C_{k_2 k_1}$, which leads to $C_{k_1 k_2} = C_{k_2 k_1} = 0$. Therefore, $\forall l \in M, \sum_{i \neq \hat{y},l} \exp(C_{il})$ has the same value.

This equation consequently results in

$$
\begin{aligned}
\arg\min_{z_0} (\max_{z^* \in Q_0^{\hat{y}}} F(z^*)_{\hat{y}}) &= \arg\min_{z_0} \frac{1}{2 + \sum_{i \neq \hat{y},\widetilde{k}} \exp(C_{i\widetilde{k}})} \\
&= \arg\max_{z_0} \sum_{i \neq \hat{y},\widetilde{k}} \exp(C_{i\widetilde{k}}).
\end{aligned}
$$

From the conclusion in Lemma 1, we know that the value $\sum_{i \neq \hat{y},\widetilde{k}} \exp(C_{i\widetilde{k}})$ obtains its maximum when $C_{i\widetilde{k}} = 0, \forall i \neq \hat{y}, \widetilde{k}$. Thus the solution set of the above problem is $S_{\hat{y}}$. Furthermore, we have $\forall z^* \in S_{\hat{y}} \bigcap \overline{dd_{\hat{y}}}, F(z^*)_{\hat{y}} = \frac{1}{2+L-2} = \frac{1}{L}$.

$\square$

**Theorem 2.** *Let $(x, y)$ be a given training data. Under the $L_\infty$-norm, if there is a training error $\alpha \ll \frac{1}{L}$ that $\|\mathbb{S}(Z_{pre}(x, \theta_R^*)) - R_y\|_\infty \le \alpha$, then we have bounds*

$$\|\mathbb{S}(-Z_{pre}(x, \theta_R^*)) - 1_y\|_\infty \le \alpha(L-1)^2,$$

*and $\forall j, k \ne y$,*

$$|\mathbb{S}(-Z_{pre}(x, \theta_R^*))_j - \mathbb{S}(-Z_{pre}(x, \theta_R^*))_k| \le 2\alpha^2(L-1)^2.$$

*Proof.* For simplicity we omit the dependence of the logits $Z_{pre}$ on the input $x$ and the parameters $\theta_R^*$. Let $G = (g_1, g_2, ..., g_L)$ be the exponential logits, where $g_i = \exp(Z_{pre,i})$. Then from the condition $\|\mathbb{S}(Z_{pre}) - R_y\|_\infty \le \alpha$ we have

$$\begin{cases} \frac{g_y}{\sum_i g_i} \le \alpha \\ \left| \frac{g_j}{\sum_i g_i} - \frac{1}{L-1} \right| \le \alpha \quad j \ne y. \end{cases}$$

Let $C = \sum_i g_i$, we can further write the condition as

$$\begin{cases} g_y \le \alpha C \\ (\frac{1}{L-1} - \alpha)C \le g_j \le (\frac{1}{L-1} + \alpha)C \quad j \ne y. \end{cases}$$

Then we can have bounds ($L \ge 2$)

$$\begin{aligned} \mathbb{S}(-Z_{pre})_y &= \frac{\frac{1}{g_y}}{\frac{1}{g_y} + \sum_{i \ne y} \frac{1}{g_i}} \\ &= \frac{1}{1 + \sum_{i \ne y} \frac{g_y}{g_i}} \\ &\ge \frac{1}{1 + \sum_{i \ne y} \frac{\alpha C}{(\frac{1}{L-1} - \alpha)C}} \\ &= \frac{1}{1 + \frac{\alpha(L-1)^2}{1 - \alpha(L-1)}} \\ &= 1 - \frac{\alpha(L-1)^2}{1 - \alpha(L-1) + \alpha(L-1)^2} \\ &\ge 1 - \alpha(L-1)^2 \end{aligned}$$

and $\forall j \ne y$,

$$\begin{aligned} \mathbb{S}(-Z_{pre})_j &= \frac{\frac{1}{g_j}}{\frac{1}{g_y} + \sum_{i \ne y} \frac{1}{g_i}} \\ &= \frac{\frac{g_y}{g_j}}{1 + \frac{g_y}{g_j} + \sum_{i \ne y,j} \frac{g_y}{g_i}} \\ &\le \frac{\frac{g_y}{g_j}}{1 + \frac{g_y}{g_j}} \\ &= \frac{1}{1 + \frac{g_j}{g_y}} \\ &\le \frac{1}{1 + \frac{(\frac{1}{L-1} - \alpha)C}{\alpha C}} \\ &= \alpha(L-1) \\ &\le \alpha(L-1)^2. \end{aligned}$$

Then we have proven that $\|\mathbb{S}(-Z_{pre}) - 1_y\|_\infty \leq \alpha(L-1)^2$. Furthermore, we have $\forall j, k \neq y$,

$$
\begin{aligned}
|\mathbb{S}(-Z_{pre})_j - \mathbb{S}(-Z_{pre})_k| &= \frac{\left|\frac{1}{g_j} - \frac{1}{g_k}\right|}{\frac{1}{g_y} + \sum_{i \neq y} \frac{1}{g_i}} \\
&\leq \frac{\frac{1}{(\frac{1}{L-1}-\alpha)C} - \frac{1}{(\frac{1}{L-1}+\alpha)C}}{\frac{1}{\alpha C} + \sum_{i \neq y} \frac{1}{(\frac{1}{L-1}+\alpha)C}} \\
&= \frac{\frac{L-1}{1-\alpha(L-1)} - \frac{L-1}{1+\alpha(L-1)}}{\frac{1}{\alpha} + \frac{(L-1)^2}{1+\alpha(L-1)}} \\
&= \frac{2\alpha^2(L-1)^2}{1 + \alpha(L-1)^2(1-\alpha L)} \\
&\leq 2\alpha^2(L-1)^2.
\end{aligned}
$$

$\square$

# B Additional Experiments

## B.1 Training Settings

We apply the same hyperparameters when training Resnet networks via the CE and RCE. The optimizer is SGD with momentum, and the mini-batch size is 128. The weight decay is 0.0002, the leakiness of Relu is 0.1.

On MNIST the training steps are 20,000, with piecewise learning rate as

$$\text{steps:}[10,000, 15,000, 20,000],$$

$$\text{lr:}[0.1, 0.01, 0.001, 0.0001].$$

Each training image pixel values are scaled to be in the interval $[-0.5, 0.5]$.

On CIFAR-10 the training steps are 90,000, with piecewise learning rate as

$$\text{steps:}[40,000, 60,000, 80,000],$$

$$\text{lr:}[0.1, 0.01, 0.001, 0.0001].$$

The training set is augmented by two ways as

- Resizing images to $40 \times 40 \times 3$ and then randomly cropping them back to $32 \times 32 \times 3$.

- Randomly flipping images along their second dimension, which is width.

After augmentation each training image pixel values are also scaled to be in the interval $[-0.5, 0.5]$.

## B.2 Time Costs on Crafting Adversarial Examples

Our experiments are done on NVIDIA Tesla P100 GPUs. We set the binary search steps to be 9 and the maximal iteration steps to be 10,000 in C&W-family attacks (i.e., C&W, C&W-hc and C&W-wb), which promises large enough searching capacity for these attacks. We set the maximal iteration steps to be 100 for JSMA, which means that JSMA perturbs at most 100 pixels on each image. Table 1 demonstrates the average time costs on crafting each adversarial example via different attacks. We can find that C&W-family attacks are extremely time consuming compared to other iterative methods. Furthermore, C&W-family attacks usually take longer time to attack the networks trained by the RCE than those trained by the CE.

Table 1: The average time costs (s) on crafting each adversarial example via different attacks. The values are also the average values between MNIST and CIFAR-10. The models is Resnet-32.

| Attack | Objective | Time |
|---|---|---|
| FGSM | CE | $\sim 1.9 \times 10^{-3}$ |
|  | RCE | $\sim 2.4 \times 10^{-3}$ |
| BIM | CE | $\sim 3.3 \times 10^{-3}$ |
|  | RCE | $\sim 3.6 \times 10^{-3}$ |
| ILCM | CE | $\sim 4.1 \times 10^{-3}$ |
|  | RCE | $\sim 4.3 \times 10^{-3}$ |
| JSMA | CE | $\sim 2.9 \times 10^{1}$ |
|  | RCE | $\sim 2.0 \times 10^{1}$ |
| C&W | CE | $\sim 4.5 \times 10^{1}$ |
|  | RCE | $\sim 5.5 \times 10^{1}$ |
| C&W-hc | CE | $\sim 6.5 \times 10^{1}$ |
|  | RCE | $\sim 1.1 \times 10^{2}$ |
| C&W-wb | CE | $\sim 7.0 \times 10^{2}$ |
|  | RCE | $\sim 1.3 \times 10^{3}$ |

## B.3 Robustness to Noisy Examples

For more complete analysis, we investigate whether our method can distinguish between noisy examples and adversarial examples. The noisy examples (RAND) here are defined as

$$x^* = x + U(-\epsilon, \epsilon)$$

where $U(-\epsilon, \epsilon)$ denotes an element-wise distribution on the interval $[-\epsilon, \epsilon]$. Fig. 1 gives the classification error rates on the test set of CIFAR-10, where $\epsilon_{RAND} = 0.04$. We find that the networks trained by both the CE and RCE are robust to noisy examples in the sense of having low error rates.

Figure 1: Classification error rates on CIFAR-10. Two panels separately show the results when the networks are trained via the CE and RCE. The models is Resnet-32.

Furthermore, in Fig. 2 and Fig. 3, we show the number of images w.r.t. the values of K-density under various attacks, also on normal and noisy examples. We work on 1,000 test images of CIFAR-10, and our baseline is the kernel density estimate method (CE as the objective and K-density as the metric). We can see that the baseline returns quite different distributions on K-density between normal and noisy examples, and it cannot distinguish noisy examples from the adversarial ones crafted by, e.g., JSMA and C&W-hc, as shown in Fig. 2. In comparison, our method (RCE as the objective and K-density as the metric) returns similar distributions on K-density between normal and noisy examples, and noisy examples can be easily distinguished from other adversarial ones, as shown in Fig. 3.

## B.4 The Limitation of C&W-wb

When we apply the C&W-wb attack, the parameter $\kappa$ is set to be 0. This makes C&W-wb succeed to fool the K-density detector but fail to fool the confidence metric. Thus we construct a high-confidence version of C&W-wb, where we set $\kappa$ be 5. However, none of the crafted adversarial examples can have $f_2(x^*) \leq 0$, as shown in Table 2. This means that it is difficult for C&W-wb to simultaneously fool both the confidence and the K-density metrics.

Table 2: The ratios (%) of $f_2(x^*) > 0$ of the adversarial examples crafted by the high-confidence version of C&W-wb on MNIST and CIFAR-10. The model is Resnet-32 and the metric is K-density.

| Objective | MNIST | CIFAR-10 |
|-----------|-------|----------|
| CE | 100 | 100 |
| RCE | 100 | 100 |

Figure 2: Number of images w.r.t. K-density. The target networks are trained by the CE.

Figure 3: Number of images w.r.t. K-density. The target networks are trained by the RCE.

## B.5 Extended experiments

Usually there is a binary search mechanism of the parameter $c$ in C&W attacks to obtain minimal adversarial perturbation. In Fig. 4 we show the extended experiment result of classification accuracy under C&W attacks with different values of $c$.

Figure 4: The network is Resnet-32, the dataset is CIFAR-10.