[Reviews · NeurIPS 2018]

Reviewer 1



This paper proposes a combination of two modifications to make neural networks robust to adversarial examples: (1) reverse cross-entropy training allows the neural network to learn to better estimate its confidence in the output, as opposed to standard cross-entropy training, and (2) a kernel-density based detector detects whether or not the input appears to be adversarial, and rejects the inputs that appear adversarial. The authors appear to perform a proper evaluation of their defense, and argue that it is robust to the attacker who performs a white-box evaluation and optimizes for evading the defense. The defense does not claim to perfectly solve the problem of adversarial examples, but the results appear to be correctly verified. As shown in Figure 3, the adversarial examples on the proposed defense are visually distinguishable from the clean images. It is slightly unclear what is meant by "ratio" in Table 3. In particular, what is a ratio of 1 or 0? Are these values rounded from something different, or are they actually 1 and 0? I am left with one question after reading this defense: what fraction of the difficulty to evade this defense comes from evading the KD detection, and what fraction comes from using the RCE training? It would be interesting to perform a study where KD is used without RCE, and RCE without KD, to determine the relative contribution of both. Does each piece require the other to be useful, or does one piece contribute more towards the defense? Minor comments: - Are the results of Table 1 statistically significant? The accuracies are very close, and so it may or may not be the case that RCE actually gives a lower misclassification rate. - If space is needed, Section 4.3 can be cut in its entirety. The problem of detecting attacks that are not attempting to evade the defense is uninteresting and has been solved several times before - The authors may want to cite Feynman et al. 2017 who also employ a KD-based adversarial example detector on the final hidden layer. Are there any important differences between this work and that work (other than RCE training)?

Reviewer 2



Update: Thank you for addressing my comments. Please make sure to update your paper according to your responses (in particular about adding additional experimental results in the appendix and updating the current set of results). In this paper, the authors propose a new loss (i.e., reverse cross-entropy) whose goal is to force non-maximal probability predictions to be uniform (in a classification task). In other words, the goal of the classification network is to maximize the entropy of all non-maximal softmax outputs. The authors show that the intermediate representations learned by the network are more meaningful when it comes to compare real and adversarial examples. Experiments are done on MNIST and CIFAR-10. The paper is well written and clearly explained. The proposed loss is analyzed both in practice and in theory. I am a bit concerned with novelty as the "reverse cross-entropy" loss is not strictly new (as pointed out by the authors themselves in Section 3.2). However, the added analysis and explanation w.r.t. the use-case of detecting adversarial examples is valuable to the community. The authors provide enough evidence that this loss is useful and, as such, I am leaning towards acceptance. The experimental section would benefit from a few added experiments. 1) In general, the problem of detecting adversarial examples is important. However, the proposed solutions are not yet compelling. The authors explain that "in semi-autonomous systems, the detection of adversarial examples would allow disabling autonomous operation and requesting human intervention", but the detection method is often easier to attack - and keeping the system from being used can also have serious consequences. The authors should elaborate on this problem and also include experiments that only aim to disable the classifiers. In general the AUC of the detector remains poor and I am assuming that a lot of non-adversarial examples are still detected as adversarial. Details: a) There has been a lot of exciting and new work on verification and training provably robust networks (see Wong & Kolter arXiv:1805.12514, arXiv:1711.00851 and Krishnamurthy et al. arXiv:1805.10265, arXiv:1803.06567). They are definitely orthogonal approaches to preventing misclassification of adversarial perturbations, but I feel they should find a space in your introduction. b) As previously stated, it would useful to incorporate results about the minimal detectable perturbation radius. c) Fig 2(a) should include Carlini & Wagner, I am unsure why a separate plot is needed. Similarly, the analysis done in Fig. 2(b) could be extended to include the other attacks.

Reviewer 3



The work proposes to combine Reverse Cross-Entropy (RCE) training and K-density detector for robust adversarial example detection. RCE encourages a deep network to learn latent representations that better distinguish adversarial examples from normal ones. The method was validated on the MNIST and CIFAR-10 datasets. Pros: 1. A metric for assessing whether an input is adversarial is proposed. 2. The effectiveness of reverse cross-entropy training is justified both theoretically and empirically. 3. The authors showed the advantages of the proposed method when handling oblivious, white-box and black-box adversaries, through experiments. Cons: 1. To better prove the effectiveness of the proposed method, the authors should evaluate it on the larger ImageNet dataset, in addition to MNIST and CIFAR-10.